# Deciphering the complex relationship between type 2 diabetes and fracture risk with both genetic and observational evidence

**Pianpian Zhao[1,2,3,4†], Zhifeng Sheng[5†], Lin Xu[6†], Peng Li[7], Wenjin Xiao[8], Chengda Yuan[9], Zhanwei Xu[10], Mengyuan Yang[1,3,4], Yu Qian[1,3,4], Jiadong Zhong[1,3,4], Jiaxuan Gu[1,3,4], David Karasik[11], Hou-Feng Zheng[1,2,3,4]\***

[1]The affiliated Hangzhou first people's hospital, School of Medicine, Westlake University, Hangzhou, China; [2]Diseases & Population (DaP) Geninfo Lab, School of Life Sciences, Westlake University, Hangzhou, China, Hangzhou, China; [3]Westlake Laboratory of Life Sciences and Biomedicine, Hangzhou, China; [4]Institute of Basic Medical Sciences, Westlake Institute for Advanced Study, Hangzhou, China; [5]Health Management Center, The Second Xiangya Hospital of Central South University, Changsha, China; [6]Department of Orthopedics, Yantai Affiliated Hospital of Binzhou Medical University, Yantai, China; [7]Department of Geratology, The Third People's Hospital of Hangzhou, Hangzhou, China; [8]Department of Endocrinology, Second Affiliated Hospital of Soochow University, Suzhou, China; [9]Department of Dermatology, Hangzhou Hospital of Traditional Chinese Medicine, Hangzhou, China; [10]Central Health Center of Mashenqiao Town, Tianjin, China; [11]Azrieli Faculty of Medicine, Bar-Ilan University, Safed, Israel

**\*For correspondence:**
zhenghoufeng@westlake.edu.cn

[†]These authors contributed equally to this work

**Abstract** The 'diabetic bone paradox' suggested that type 2 diabetes (T2D) patients would have higher areal bone mineral density (BMD) but higher fracture risk than individuals without T2D. In this study, we found that the genetically predicted T2D was associated with higher BMD and lower risk of fracture in both weighted genetic risk score (wGRS) and two-sample Mendelian randomization (MR) analyses. We also identified ten genomic loci shared between T2D and fracture, with the top signal at SNP rs4580892 in the intron of gene *RSPO3*. And the higher expression in adipose subcutaneous and higher protein level in plasma of *RSPO3* were associated with increased risk of T2D, but decreased risk of fracture. In the prospective study, T2D was observed to be associated with higher risk of fracture, but BMI mediated 30.2% of the protective effect. However, when stratified by the T2D-related risk factors for fracture, we observed that the effect of T2D on the risk of fracture decreased when the number of T2D-related risk factors decreased, and the association became non-significant if the T2D patients carried none of the risk factors. In conclusion, the genetically determined T2D might not be associated with higher risk of fracture. And the shared genetic architecture between T2D and fracture suggested a top signal around *RSPO3* gene. The observed effect size of T2D on fracture risk decreased if the T2D-related risk factors could be eliminated. Therefore, it is important to manage the complications of T2D to prevent the risk of fracture.

## eLife assessment

This study aims to explore the diabetes-bone paradox using the Mendelian Randomization approach. That diabetes itself is not the direct cause, but rather the complications or associated

risk factors increase the risk of fracture, constitutes a **valuable** insight. Mendelian randomization to explain the relationship of two complex conditions is **solid** and conducted properly; however, the efforts to reconcile the discrepancies between the Mendelian Randomization analysis and observational studies are **incomplete**.

## Introduction

Type 2 diabetes (T2D), a chronic metabolic disorder characterized by elevated blood glucose levels and increased risk of numerous serious and life-threatening complications, constitutes one of the biggest health problems in the world (*Diamond, 2003*). According to Global Burden of Disease (GBD) data, the age-standardized global prevalence of type 2 diabetes was approximately 6.0% in men and 5.0% in women in 2019 (*Tinajero and Malik, 2021*). It accounts for more than 100 billion dollars of healthcare costs annually in the United States (*Diamond, 2003*). The chronic comorbidities of T2D could develop gradually, and could lead to serious damage to the heart, blood vessels, kidneys, eyes, and feet (*Teck, 2022*). Other organ systems such as skeletal health could also be influenced by T2D (*Lei and Kindler, 2022*).

Osteoporosis is a common musculoskeletal disease characterized by low bone mass and disruption of bone microarchitecture, leading to an increased risk of fracture. Our previous studies have suggested that bone mass and fracture could be influenced by many modifiable or non-modifiable factors (*Zhu and Zheng, 2021*), such as body weight (*Zhu et al., 2022*), sleep behavior (*Qian et al., 2021*), inflammatory disease (*Xia et al., 2020*), birth weight (*Xia et al., 2022*), and genetic factors (*Zhu et al., 2021*). T2D is also considered to be a major factor that could affect bone health, it seems that T2D patients would have higher BMD and higher fracture risk than individuals without T2D (*Khosla et al., 2021*). This is the so-called 'diabetic bone paradox' (*Botella Martínez et al., 2016*; *Romero-Díaz et al., 2021*). For example, in an Italian nationwide study of 59,950 women of whom 5.2% had diabetes (predominantly type 2 diabetes), noted an association between diabetes and any fracture (OR 1.3, 95% CI 1.1–1.4, and OR 1.3, 95% CI 1.2–1.5, for vertebral or hip fractures and non-vertebral, non-hip fractures, respectively) (*Adami et al., 2020*). Interestingly, the prevalence of vertebral or hip fracture was higher in participants with diabetes but without obesity (OR 1.9, 95% CI 1.7–2.1) than in patients with obesity and diabetes (OR 1.5, 95% CI 1.3–1.8), suggesting that obesity might be partially protective against vertebral or hip fractures in type 2 diabetes (*Adami et al., 2020*).

However, a recent comparative cohort analysis using routinely collected UK primary care records data from the Health Improvement Network (including 174,244 individuals with incident type 2 diabetes and 747,290 without diabetes) found no evidence to suggest a higher risk of fracture in type 2 diabetes patients, specifically, the risk of having at least one fracture was estimated to be 6% lower for females and 3% lower for males in the type 2 diabetes cohort than for females and males without diabetes (*Davie et al., 2021*). Lower fracture risk was also observed in the type 2 diabetes patients compared to those without the disease in the age group greater than 85 years (*Davie et al., 2021*). Another large-scale cohort study showed that type 2 diabetes could only explain less than 0.1% of the fracture risk (*Axelsson et al., 2023*), and if the T2D patients with risk factors (such as low BMI, long diabetes duration, insulin treatment, and low physical activity) were excluded, T2D patients would have lower fracture risk than their matched controls (*Axelsson et al., 2023*). In a prospective study to examine the relationship between BMD and fracture in older adults with type 2 diabetes, it was reported that femoral neck BMD T score and FRAX score were both associated with fracture risk in individuals with type 2 diabetes, suggesting that BMD is still a useful clinical predictor for the evaluation of fracture risk in type 2 diabetes patients (*Schwartz et al., 2011*).

As the pathophysiology of fracture is more complicated than the BMD trait, and while there were some explanations for the 'diabetic bone paradox' (*Osório, 2011*), the integrated analyses with genetic data for the diseases could provide an alternative approach to alleviate the bias of the unknown confounding factors (*Davey Smith and Hemani, 2014*). Therefore, in this study, we first performed a weighted genetic risk score (wGRS) regression analysis to assess the relationship between the genetically predicted T2D and fracture with genetic summary data and individual genotype data in UK biobank. The two-sample Mendelian randomization (MR) approach was used as an independent validation analysis. We applied the MiXeR method and conditional/conjunctional false discovery rate (ccFDR) approach to identify the shared genetic components between the traits. Finally, within

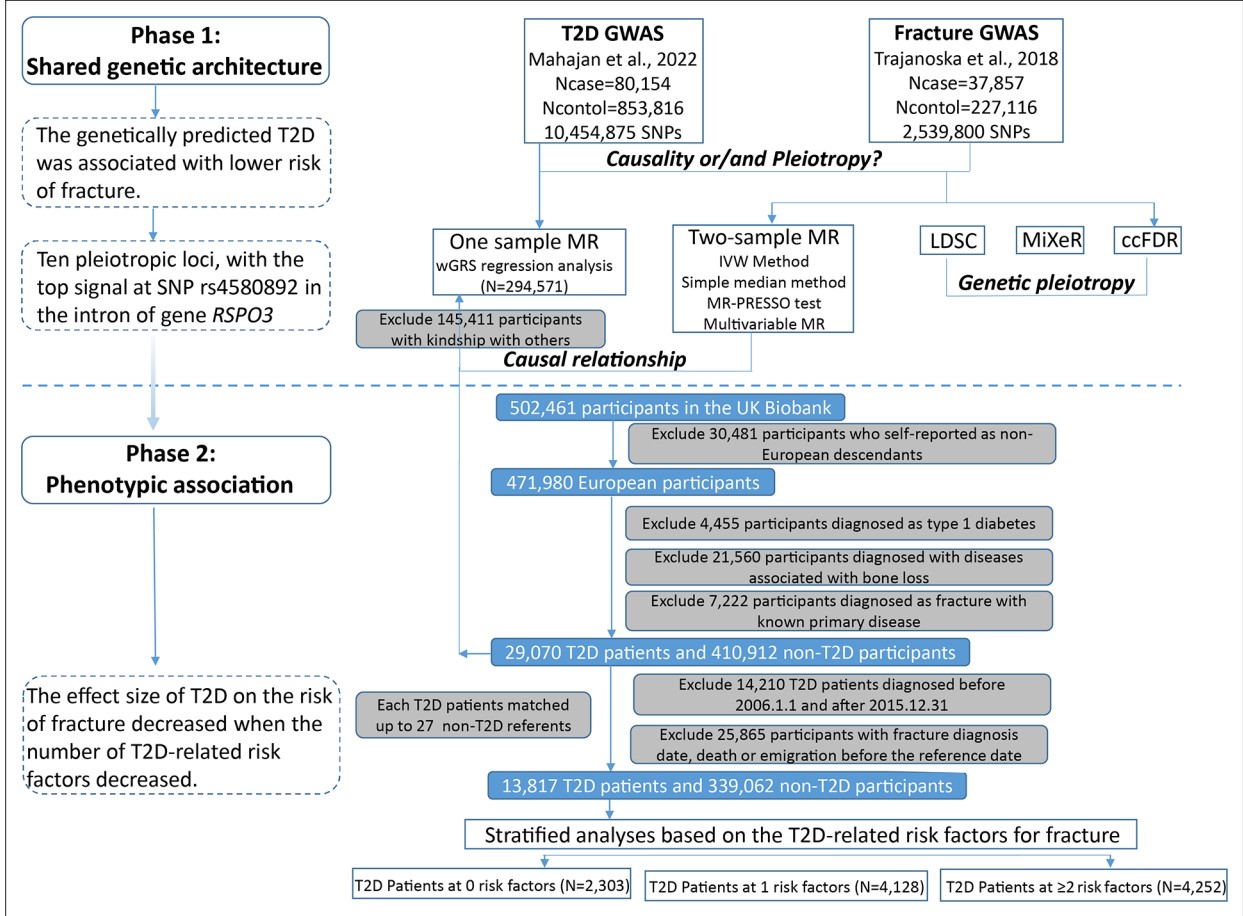

**Figure 1.** Flow chart of the overall study design.

The online version of this article includes the following figure supplement(s) for figure 1:

**Figure supplement 1.** The matching information of 14,860 patients with fractures.

the UK biobank dataset, the stratified cox regression analyses were applied to explore the association between T2D and fracture risk by including different number of the T2D-related risk factors. As complement, the relationship between T2D and BMD was also investigated.

## Results

### The genetically predicted type 2 diabetes and fracture

The overall study design was presented in *Figure 1*. We first assessed the relationship between genetically predicted T2D and fracture in theUK biobank dataset with the wGRS analysis. Within the 294,571 UK biobank samples (*Figure 1*), we constructed the wGRS for the individuals in the UK Biobank with the 404 SNPs, which were independently associated with type 2 diabetes (*Supplementary file 1a*). The wGRS of the 404 SNPs were strongly associated with type 2 diabetes in UK Biobank data (OR = 1.6, p<2.0 × 10$^{-16}$), suggesting that the instruments were powerful for the MR analysis. When we regressed the observed fracture on the wGRS, we found that the genetically determined type 2 diabetes was associated with a lower risk of fracture (OR = 0.982, 95%CI = 0.975–0.989, p=0.006) (adjusting for reference age, sex, BMI, physical activity, fall history, HbA1c, and medication treatments) (*Figure 2A*). When we classified the fracture sites into weight-bearing bones (neck, vertebrae, pelvic, femur, tibia) and other bones, it indicated that there was a trend of protective association between T2D wGRS and weight-bearing bones fracture (OR = 0.9772, 95%CI = 0.9552–0.9997, p=0.04737, N of fracture = 8992, N of non-fracture=265,262), and other bones fracture (OR = 0.9838, 95%CI = 0.9688–0.9991, p=0.0386, N of fracture = 20,317, N of non-fracture=265,262) (*Figure 2A*). We further estimated the

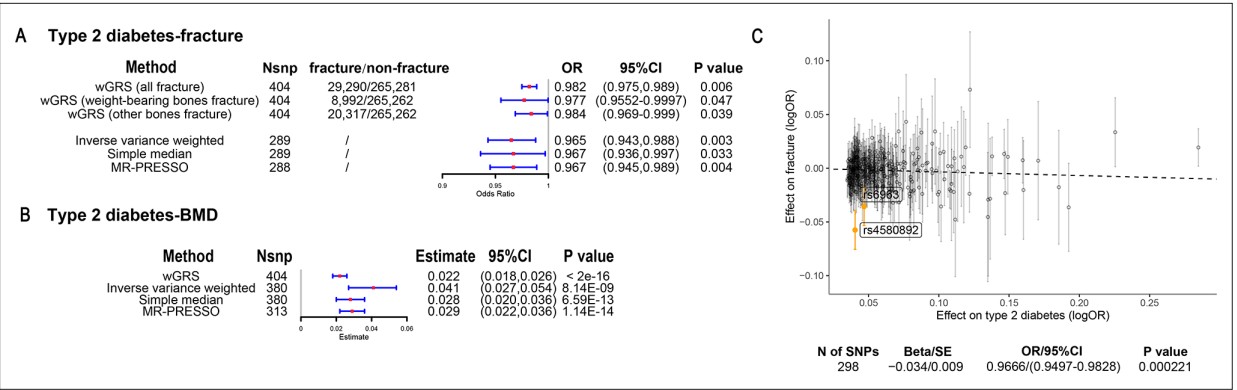

**Figure 2.** The association of genetically predicted type 2 diabetes with fracture and bone mineral density (BMD) by using different Mendelian randomization (MR methods). (**A**) The genetically predicted type 2 diabetes and fracture. (**B**) The genetically predicted type 2 diabetes and BMD. (**C**) Visualized the association of lead SNPs for type 2 diabetes with the risk of fracture. Abbreviations: wGRS, weighted genetic risk score; MR-PRESSO, MR pleiotropy residual sum and outlier.

The online version of this article includes the following figure supplement(s) for figure 2:

**Figure supplement 1.** The regression between type 2 diabetes GRS and fracture risk/bone mineral density (BMD) in male and female.

effect of sex interaction on fracture risk with T2D *wGRS × sex* interaction term in the regression model, and no significant interactions were identified for fracture risk (p=0.5576). Moreover, we conducted the stratified analysis by sex and identified similar trends of association (*Figure 2—figure supplement 1A*). Meanwhile, the genetically determined type 2 diabetes was associated with higher BMD in pooled samples (*Figure 2B*) and in both males and females (*Figure 2—figure supplement 1B*).

We also performed the two-sample MR analyses with fracture GWAS summary data (*Trajanoska et al., 2018*) which is independent of UK Biobank samples. The inverse variance weighting (IVW) method showed a causal effect of genetically predicted T2D on low fracture risk (OR = 0.965, 95%CI = 0.943–0.988, p=0.003) (*Figure 2A*) using 298 SNPs as the instruments (*Supplementary file 1b*). This causal relationship was also significant in the simple median test (OR = 0.967, 95%CI = 0.936–0.997, p=0.033) (*Figure 2A*). There was heterogeneity in IVW results (Q' p<0.05), when we excluded pleiotropic variants using restrictive MR pleiotropy residual sum and outlier test (MR-PRESSO) method, the causal association was still detected between T2D and fracture (OR = 0.967, 95%CI = 0.945–0.989, p=0.004) (*Figure 2A*). Moreover, the MR-egger regression also suggested an inverse association between T2D and fracture (OR = 0.9666, 95%CI = 0.9497–0.9828, p=0.0002) (*Figure 2C*). The individual effect of the SNPs for T2D on fracture was corrected by the false discovery rate (<0.05) (*Benjamini and Hochberg, 1995*), two of 298 lead SNPs (including rs4580892 near *RSPO3*) of T2D remained as potential regions which would also have effect on fracture (*Figure 2C* and *Supplementary file 1b*). We also performed multivariable MR analysis to test the effect of T2D on fracture risk-adjusted for confounding factors. We found that T2D had a direct effect on decreased fracture risk adjusted for BMI (OR = 0.974, 95%CI = 0.953–0.995, p=0.017), and BMI mediated 9.03% of the protective effect (*Supplementary file 1c*). Similarly, with BMD GWAS summary data (*Morris et al., 2019*), IVW ($\beta$=0.041, 95%CI = 0.027–0.054, p=8.14 × 10⁻⁹), simple median ($\beta$=0.028, 95%CI = 0.020–0.036, p=6.59 × 10⁻¹³), and MR-PRESSO ($\beta$=0.029, 95%CI = 0.022–0.036, p=1.14 × 10⁻¹⁴) all showed a causal association between type 2 diabetes and BMD (*Figure 2B*, *Supplementary file 1d*). The multivariable MR analysis suggested that T2D also showed a direct effect on increased BMD after adjusting for BMI ($\beta$=0.042, 95%CI = 0.026–0.057, p=1.92 × 10⁻⁷) (*Supplementary file 1c*).

## The distinct signal shared by type 2 diabetes and fracture

Leveraging the genetic summary datasets, we first evaluated the genetic correlation among the traits and diseases by using LDSC (*Bulik-Sullivan et al., 2015a*). It is found that the genetic correlation between type 2 diabetes and fracture was not significant, but with inverse direction ($r_g$ = −0.0114) (*Supplementary file 1e*). Instead, we used MiXeR (*Frei et al., 2019a*; *Frei et al., 2019b*) to evaluate the polygenic overlap irrespective of the genetic correlation between T2D and fracture. As represented in Venn diagrams of shared and unique polygenic components (*Figure 3a*), the MiXeR analysis

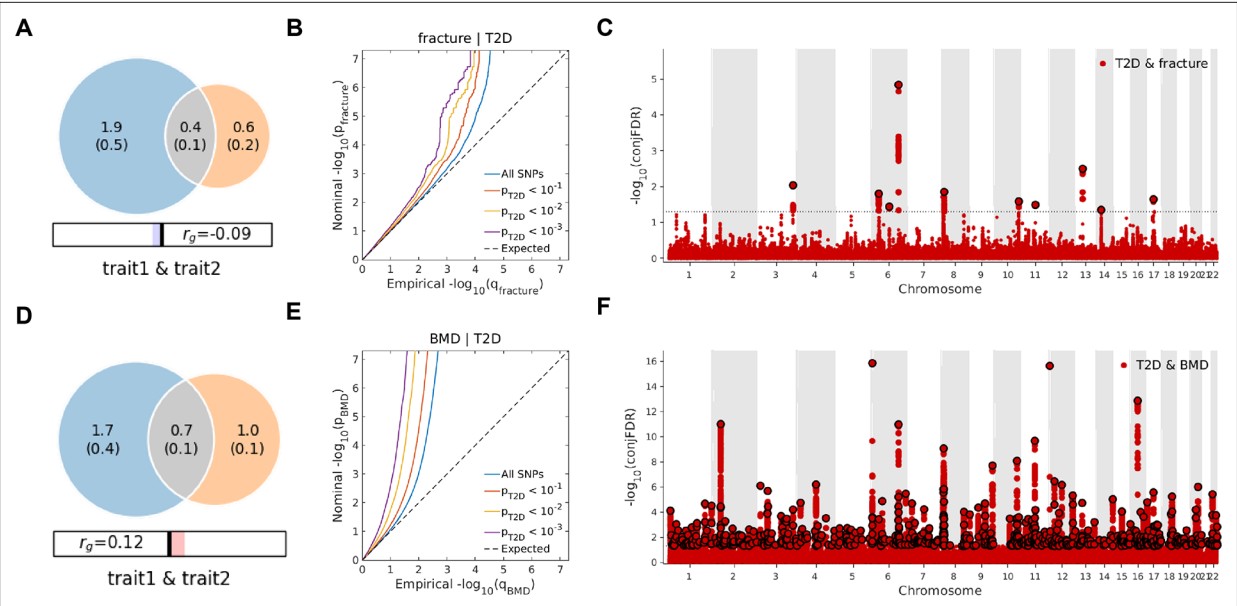

**Figure 3.** The Venn diagrams, conditional quantile-quantile (QQ) plots, and genetic variants jointly associated with type 2 diabetes and fracture/bone mineral density (BMD) at conjunctional false discovery rate (conjFDR) <0.05. (**A**) The shared number of variants between type 2 diabetes and fracture. (**B**) The conditional QQ plot of fracture given the association with type 2 diabetes at the level of p≤0.1, p≤0.01, p≤0.001. (**C**) The shared genetic loci between type 2 diabetes and fracture. (**D**) The shared number of variants between type 2 diabetes and BMD. (**E**) The conditional QQ plot of BMD given the association with type 2 diabetes at the level of p≤0.1, p≤0.01, p≤0.001. (**F**) The shared genetic loci between type 2 diabetes and BMD.

suggested that type 2 diabetes and fracture exhibited polygenic overlap, sharing 428 causal variants, in other words, 18% of variants (428 of 2370) associated with type 2 diabetes might contribute to the risk of fracture (Dice coefficient = 25.25%), and genetic correlation was observed ($r_g$ = −0.086) (**Figure 3A** and **Supplementary file 1f**). Only 39% of shared variants between type 2 diabetes and fracture showed a concordant direction of the association, and the correlation of effect sizes within the shared polygenic component was negative (rho_$\beta$ = −0.336) (**Supplementary file 1f**).

We used the ccFDR approach (**Andreassen et al., 2013b**) to identify specific shared loci between type 2 diabetes and fracture from the GWAS summary statistics. The stratified conditional QQ plot was utilized to visualize the enrichment of association with fracture across varying significance thresholds for type 2 diabetes. We observed leftward deflected from the expected null line in QQ plot, which suggested the existence of a polygenic overlap between type 2 diabetes and fracture (**Figure 3B**).

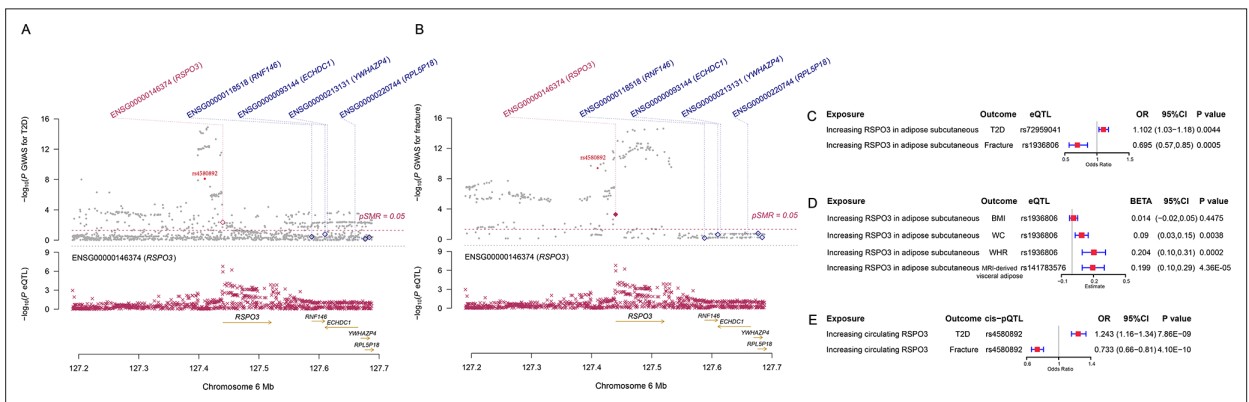

**Figure 4.** The distinct signal (*RSPO3*) shared by type 2 diabetes and fracture. (**A**) The regional plot of the association of type 2 diabetes and *RSPO3* gene expression (adipose subcutaneous) within hg19: chr6:127189749–127689749 (*RSPO3* gene region ±250 kb window). (**B**) The regional plot of the association of fracture and *RSPO3* gene expression (adipose subcutaneous) in the same region. (**C**) The association of genetically predicted *RSPO3* gene expression with type 2 diabetes and fracture risk. (**D**) The association of genetically predicted *RSPO3* gene expression with BMI, waist circumference, waist-hip ratio, and MRI-derived visceral adipose. (**E**) The association of genetically predicted circulating RSPO3 with type 2 diabetes and fracture risk.

The conjunctional false discovery rate (conjFDR) analysis identified 10 genomic loci shared between type 2 diabetes and fracture (*Figure 3C* and *Supplementary file 1g*), with the top SNP rs4580892 in the intron of gene *RSPO3* (conjFDR = 1.45E-05). The shared loci showed mixed directions of allelic associations, with 7 of 10 shared loci had an inverse direction of effect between type 2 diabetes and fracture (*Supplementary file 1g*). We found that the locus approximately 250 kb upstream and downstream of the gene *RSPO3* (hg19, chr6: 127189749–127689749) possessed many significant SNPs associated with type 2 diabetes and fracture (*Figure 4A and B*), with the nearest gene *RSPO3*. The top SNP rs4580892 had an inverse direction of effect between type 2 diabetes and fracture, where rs4580892_T allele was associated with increased type 2 diabetes risk (OR = 1.041227, p=$8.46 \times 10^{-9}$) (*Figure 4A*) and decreased fracture risk (OR = 0.944, p=$3.68 \times 10^{-10}$) (*Figure 4B*).

Furthermore, after merging the eQTL summary data of *RSPO3* in adipose subcutaneous with the summary data of type 2 diabetes and fracture, rs72959041 and rs1936806 were the top cis-eQTL for type 2 diabetes and fracture in GTEx database, respectively. By applying the SMR method (*Zhu et al., 2016*), we found that the higher expression of *RSPO3* (ENSG00000146374) in adipose subcutaneous would be associated with increased the risk of type 2 diabetes (OR = 1.102, 95%CI = 1.031–1.179, p=0.004), but decreased the risk of fracture (OR = 0.695, 95%CI = 0.566–0.854, p=0.0005) (*Figure 4C* and *Supplementary file 1h*). Interestingly, in adipose subcutaneous, higher expression of *RSPO3* was associated with higher waist circumference ($\beta$=0.090, 95%CI = 0.029–0.151, p=0.004) and higher waist-hip ratio ($\beta$=0.204, 95%CI = 0.095–0.313, p=0.0002) (*Figure 4D* and *Supplementary file 1h*). Meanwhile, higher expression of *RSPO3* was associated with higher MRI-derived visceral adipose ($\beta$=0.199, 95%CI = 0.103–0.294, p=$4.36 \times 10^{-5}$) (*Figure 4D* and *Supplementary file 1h*). The association between the expression of *RSPO3* and BMI was not significant, but the direction is the same as the waist circumference (*Figure 4D* and *Supplementary file 1h*). Moreover, to estimate the impact of RSPO3 protein level on type 2 diabetes and fracture risk, we used the top SNP at rs4580892, a cis-pQTL for circulating RSPO3 (p=$2.34 \times 10^{-11}$) identified by Sun et al in an independent dataset (*Sun et al., 2018*), to instrument the circulating protein level of RSPO3. The MR analyses indicated that increased circulating RSPO3 was strongly associated with increased risk of type 2 diabetes (OR = 1.24, 95%CI = 1.16–1.34, p=$7.86 \times 10^{-9}$), but reduced fracture risk (OR = 0.73, 95%CI = 0.66–0.81, p=$4.1 \times 10^{-10}$) (*Figure 4E*).

Not surprisingly, type 2 diabetes showed a significant positive genetic correlation with BMD ($r_g$ = 0.0923, p=$2.50 \times 10^{-6}$) (*Supplementary file 1e*). The MiXeR analysis suggested that 29% of variants (691 of 2370) associated with type 2 diabetes might contribute to BMD (Dice coefficient = 33.67%) (*Figure 3D* and *Supplementary file 1f*). The leftward deflected from the expected null line in the QQ plot suggested the existence of a polygenic overlap between type 2 diabetes and BMD (*Figure 3E*). and the conjFDR analysis identified 661 genomic loci shared between type 2 diabetes and BMD, and 449 of 661 loci (68%) had concordant associations between type 2 diabetes and BMD (*Figure 3F* and *Supplementary file 1i*).

## Observed relationship between type 2 diabetes and fracture

Within the 352,879 UK Biobank participants (*Figure 1*), 13,817 (3.92%) developed type 2 diabetes during 2006 and 2015, with the mean duration of type 2 diabetes 8.34 years (*Supplementary file 1j*). Compared to those without diabetes, the participants with type 2 diabetes were older (63.20 vs 60.55, p<$2.2 \times 10^{-16}$), and more likely to be men and smokers, and had a higher BMI (32.07 vs 27.08, p<$2.2 \times 10^{-16}$) (*Supplementary file 1j*). We identified 16,147 (4.6%) participants with fractures within the 352,879 UK Biobank participants (*Supplementary file 1j*).

Although we found that genetically predicted type 2 diabetes might not be associated with risk of fracture, we observed a higher risk of fracture in the type 2 diabetes patients in the Cox proportional hazards regression after adjusting for the reference age, sex, BMI, physical activity, HbA1c, medication treatments, and fall history (Model 0) (HR = 1.527, 95% CI 1.385–1.685, p<$2 \times 10^{-16}$) (*Figure 5A* and *Supplementary file 1k*). And the average causal mediation effect (ACME) by BMI was protective with 30.2% of the intermediary effect, respectively (BMI: indirect effect = −0.003, p<$2 \times 10^{-16}$) (*Supplementary file 1l*). Similar findings were observed for both males and females (HR = 1.587, 95% CI 1.379–1.828, p=$1.26 \times 10^{-10}$ in male, HR = 1.530, 95% CI 1.334–1.756, p=$1.27 \times 10^{-9}$ in female) (*Figure 5—figure supplement 1A*). When we additionally controlled for BMD, we still observed an increased risk of fracture in type 2 diabetes (HR = 1.574, 95% CI 1.425–1.739, p<$2 \times 10^{-16}$) (Model

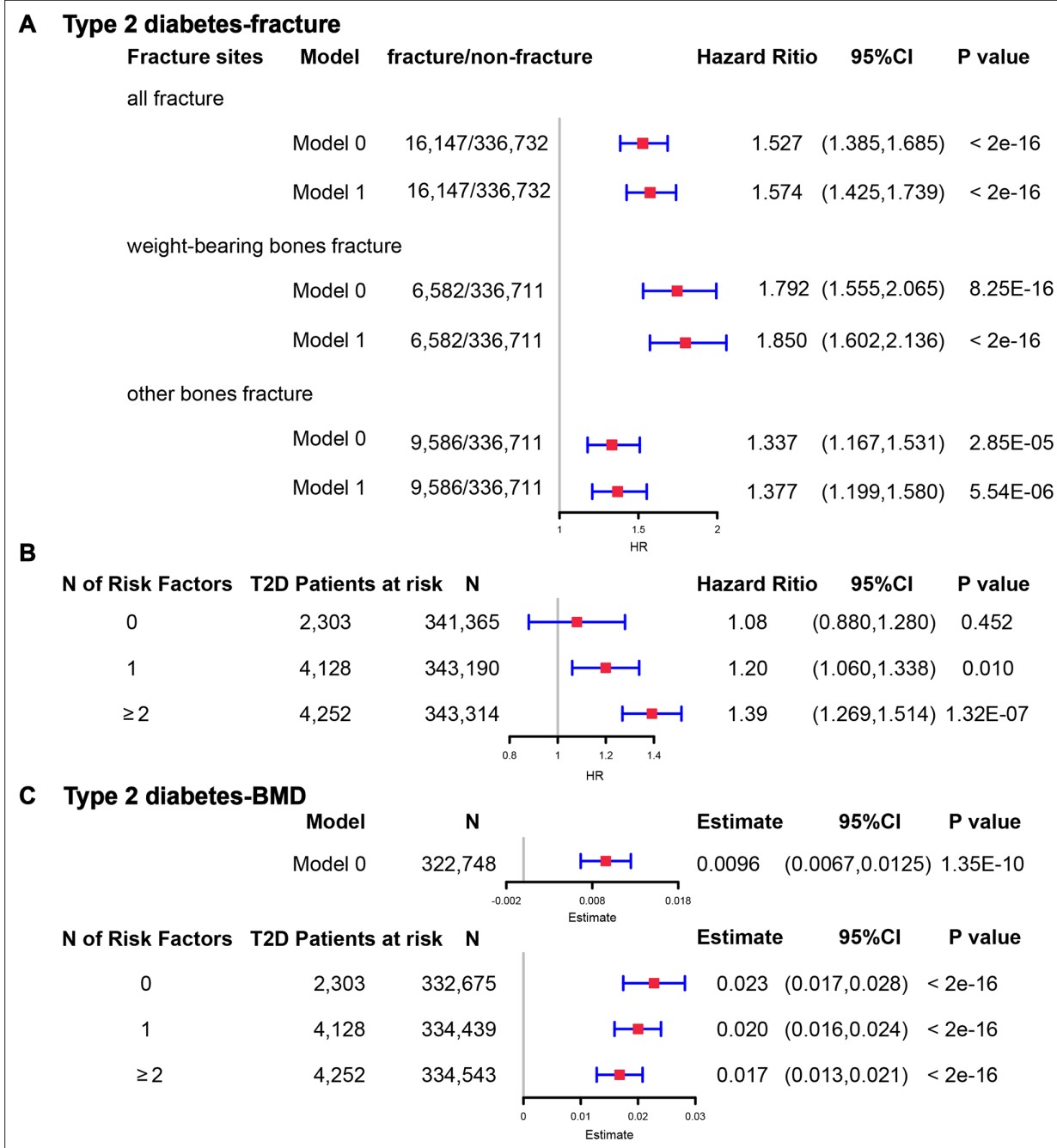

**Figure 5.** The regression between type 2 diabetes and fracture/bone mineral density (BMD) in the observational study. (**A**) The relationship between type 2 diabetes and fracture in different sites including all fractures, weight-bearing bones, and other bones. Model 0 adjusted for reference age, sex, BMI, physical activity, fall history, HbA1c, and medication treatments; Model 1 adjusted for a+ BMD. (**B**) The stratified analyses between type 2 diabetes, and fracture based on the five T2D-related risk factors for fracture adjusted for the age and sex. (**C**) The relationship between type 2 diabetes and BMD and the stratified analyses based on the five T2D-related risk factors for fracture adjusted for the age and sex. The five risk factors were: (1) BMI ≤25 kg/m²; (2) no physical activity; (3) falls in the last year; (4) HbA1c≥47.5 mmol/mol; (5) antidiabetic medication treatments.

The online version of this article includes the following figure supplement(s) for figure 5:

**Figure supplement 1.** The regression between type 2 diabetes and fracture risk/bone mineral density (BMD) in male and female in the observational study.

**Figure supplement 2.** The stratified analyses between type 2 diabetes and fracture/bone mineral density (BMD) based on the five T2D-related risk factors for fracture (BMI ≤25 kg/m², no physical activity, falls in the last year, HbA1c≥47.5 mmol/mol and antidiabetic medication treatments) in male and female.

1) (*Figure 5A* and *Supplementary file 1k*). We also classified the fracture into weight-bearing bone fractures (neck, vertebrae, pelvic, femur, tibia) and other bones fracture. Similar trends of association were observed in model 0 (weight-bearing bones: HR = 1.792, 95% CI 1.555–2,065, p=8.25 × 10⁻¹⁶; other bones: HR = 1.337, 95% CI 1.167–1.531, p=2.85 × 10⁻⁵) and model 1 (weight-bearing bones: HR = 1.850, 95% CI 1.602–2,136, p<2 × 10⁻¹⁶; other bones: HR = 1.377, 95% CI 1.199–1.580, p=5.54 × 10⁻⁶) (*Figure 5A*).

Inspired by the MR analyses that genetically determined T2D might not be a risk factor for fracture, we conducted stratified analyses based on the T2D-related risk factors for fracture, such as BMI ≤25 kg/m², no physical activity, falls in the last year, HbA1c≥47.5 mmol/mol and antidiabetic medication treatments. Within the 13,817 individuals with type 2 diabetes, 2303 patients carried none of the above risk factors, 4128 patients accompanied with one of the risk factors, and 4252 patients carried at least two risk factors (*Supplementary file 1m*). We performed stratified cox regression analysis and found that type 2 diabetes with at least two risk factors were associated with an increase in fracture risk (HR = 1.39, 95% CI 1.269–1.514, p=1.32 × 10⁻⁷) (*Figure 5B*). It is interesting to note that including only one risk factor would attenuate the effect size of type 2 diabetes on fracture risk (HR = 1.20, 95%CI = 1.060–1.338, p=0.010). Furthermore, the association between type 2 diabetes and fracture was not significant (p=0.452) when analyzing the type 2 diabetes without risk factors (HR = 1.08, 95% CI 0.880–1.280, n=2303) (*Figure 5B*). Similar trends of association were observed in both males and females (*Figure 5—figure supplement 2A*). These results suggested that the T2D-related risk factors might contribute to the risk of fracture instead of the disease itself.

We also observed that participants with diabetes, despite they were older, had a significantly higher BMD than subjects without diabetes (0.57 vs 0.54, p<2.2 × 10⁻¹⁶) (*Supplementary file 1j*). In the multivariable linear regression analysis, the type 2 diabetes was found to be associated with increased BMD in the same model adjusted for age, sex, BMI, physical activity, fall history, HbA1c, and medication treatments (β=0.00957, p=1.35 × 10⁻¹⁰) (*Figure 5CSupplementary file 1k*). We examined the relationship between type 2 diabetes and BMD in subgroups with varying numbers of risk factors. We observed that the effect size of type 2 diabetes on BMD decreased when the number of risk factors increased (no risk factors: in pooled β=0.023, p<2 × 10⁻¹⁶, in male β=0.018, p=2.09 × 10⁻⁶, in female β=0.028, p=2.97 × 10⁻¹¹; one risk factor: in pooled β=0.020, p<2 × 10⁻¹⁶, in male β=0.015, p=1.91 × 10⁻⁷, in female β=0.0245, p=2.46 × 10⁻¹⁶; at least two risk factors: β=0.017, p<2 × 10⁻¹⁶, in male β=0.0098, p=7.74 × 10⁻⁴, in female β=0.0250, p<2 × 10⁻¹⁶) (*Figure 5C*, *Figure 5—figure supplement 2B*).

## Discussion

By leveraging the genetic datasets, we found that the genetically predicted T2D was associated with higher BMD and lower risk of fracture in both one-sample MR (with 404 IVs) and two-sample MR (with 298 IVs). We also identified ten genomic loci shared between fracture and T2D, with the top signal at SNP rs4580892 in the intron of gene *RSPO3*. And the higher expression of *RSPO3* in adipose subcutaneous was associated with an increased risk of T2D, but a decreased risk of fracture. Similarly, the increased circulating RSPO3 was strongly associated with increased risk of T2D, but reduced fracture risk. In the prospective study, T2D was observed to be associated with a higher risk of fracture, but BMI mediated 30.2% of the protective effect. However, when stratified by the T2D-related risk factors for fracture, we observed that the effect size of T2D on the risk of fracture decreased when the number of T2D-related risk factors decreased, and the association became not significant if the T2D patients carried none of the risk factors.

The 'diabetic bone paradox' suggested that T2D patients would have higher areal BMD but higher fracture risk than individuals without T2D (*Botella Martínez et al., 2016*; *Romero-Díaz et al., 2021*). Other measurements, such as trabecular bone score (*Fazullina et al., 2022*; *Ho-Pham and Nguyen, 2019*) and chest CT texture analysis (*Kim et al., 2023* ), could provide additional valuable information in the evaluation of fracture risk, especially in type 2 diabetes patients. As reviewed previously, heterogeneity could exist from study to study, and conflicting observational findings were reported (*Khosla et al., 2021*). MR could be an alternative approach to infer the relationship between exposure and outcome, as this method exploits the idea that genotypes are distributed randomly at conception, facilitating their use as instrumental variables (IV) to alleviate the bias of the unknown confounding factors (*Davey Smith and Hemani, 2014*; *Zhao et al., 2019*). (*Trajanoska et al., 2018*) assessed the

effect of 15 selected clinical risk factors on the risk of fracture by using two-sample MR analysis, and reported a non-significant relationship between type 2 diabetes and fracture risk, but the direction of effect was negative (OR = 0.99). However, only 38 SNPs were extracted as instruments from a GWAS published in 2012 (*Morris et al., 2012*). In the present study, we extracted 298 T2D-associated independent SNPs from *Mahajan et al., 2022*, which is the largest-scale GWAS meta-analysis to date published in 2022, as the IVs in two-sample MR analysis. We reported that genetically determined type 2 diabetes was associated with a lower risk of fracture, even in multivariable MR analysis adjusted for BMI. In addition, we also calculated the wGRS with 404 T2D-associated independent SNPs in the UK Biobank dataset, and performed a regression analysis of wGRS of type 2 diabetes on the fracture risk (one-sample MR). Again, we found that the genetically predicted type 2 diabetes was associated with a lower risk of fracture in one-sample MR analysis. To be noted, two-sample MR results could be served as an independent replication to the one-sample MR results, because the effects of the outcome (fracture risk) for two-sample MR were derived from *Trajanoska et al., 2018*, while the one-sample MR used the UK Biobank dataset, the study samples had no overlap. Furthermore, consistent with previous studies (*Ahmad et al., 2017*; *Mitchell et al., 2021*), the MR analysis in the present study suggested that the genetically predicted type 2 diabetes was associated with higher BMD. That is to say, by alleviating the bias of the unknown confounding factors through MR analysis, the genetically predicted type 2 diabetes did not show this bone paradox.

The genetic correlation between type 2 diabetes and fracture estimated by LDSC (*Bulik-Sullivan et al., 2015b*) was not significant. It is hard for LDSC to identify the genetic pleiotropy with mixed-effect directions, which is what usually happens between two complex traits. Therefore, in this study, we employed the MiXeR method (*Frei et al., 2019b*), which could identify the unique and shared polygenic SNPs between two traits regardless of genetic correlation. The MiXeR analysis suggested that type 2 diabetes and fracture exhibited polygenic overlap, and 61% of shared variants between type 2 diabetes and fracture showed a discordant direction of the association, and the correlation of effect sizes within the shared polygenic component was negative, suggesting an inverse genetic relationship between type 2 diabetes and fracture. Additionally, the conditional/conjunctional false discovery rate analysis (*Andreassen et al., 2013b*) suggested a top locus at 6q22 (SNP rs4580892) jointly associated with type 2 diabetes and fracture. This SNP is an intronic variant in gene *RSPO3* (ENSG00000146374). We found that higher expression of *RSPO3* in adipose subcutaneous would be associated with increased risk of type 2 diabetes, but decreased risk of fracture. RSPO3 is a known WNT-signaling modulator (*Baron and Kneissel, 2013*; *Lerner and Ohlsson, 2015*), which could bind to LRP5/6 to enhance the activity of osteoblast (*Richards et al., 2012*). Karin et al., demonstrated that *RSPO3* is expressed in osteoprogenitor cells and osteoblasts, and that osteoblast-derived RSPO3 is the principal source of RSPO3 in bone and is an important regulator of vertebral trabecular bone mass and bone strength in adult mice (*Nilsson et al., 2021*). Interestingly, we also found that higher expression of *RSPO3* was associated with higher waist circumference and higher waist-hip ratio. It was reported that RSPO3 could impact body fat distribution (*Loh et al., 2020*), and fat distribution is an independent predictor of type 2 diabetes (*Manolopoulos et al., 2010*). Therefore, we speculated that the different roles of the shared genetic components between bone metabolism and type 2 diabetes might provide one possible explanation for the inverse association pattern, and obese tendency might mediate this pattern. In fact, in UK Biobank, the participants with type 2 diabetes had a higher BMI compared to those without diabetes (31.74 vs 27.05, p<2.2 × 10$^{-16}$), and BMI mediated 30.2% of the intermediary effect between type 2 diabetes and fracture in our study.

On the contrary, in the observational study, we found that type 2 diabetes was associated with a higher risk of fracture even adjusted for a bunch of confounding factors such as the age, sex, BMI, physical activity, HbA1c, medication treatments, and fall history. When fractures were categorized into different sites (weight-bearing bones and other bones), the association between type 2 diabetes and fracture remained evident. Inspired by the MR analyses that genetically determined type 2 diabetes might not be a risk factor for fracture, we started to perform the stratified analyses based on the T2D-related risk factors. There are many secondary factors associated with type 2 diabetes that might contribute to fracture risk. Our previous study suggested that keeping a moderate-high BMI (overweight) might be of benefit to old people in terms of fracture risk (*Zhu et al., 2022*), and an intensive lifestyle intervention, such as weight loss, in T2D patients might increase fracture risk (*Johnson et al., 2017*). The hyperglycemia could cause osteocyte senescence and premature programmed cell death,

leading to decreased ability to sense and respond to mechanical stimuli such as oscillatory shear stress, ultimately contributing to skeletal fragility (*Eckhardt et al., 2020*). Another major and complicated factor that might influence the risk of fracture in T2D patients is the use of diabetes mellitus medications. For example, the use of insulin (*Napoli et al., 2014*) or thiazolidinediones (*Zhu et al., 2014*) was reported to be associated with an increased risk of fracture. In addition, the risk of falls, which might be triggered by some diabetic complications such as visual impairment and peripheral neuropathy, was suggested to increase in the patients with T2D (*Schwartz et al., 2002*). Besides, low physical activity was identified as one of the most important independent diabetes-related risk factors for fracture through Gradient Boosting Machines (*Axelsson et al., 2023*). Therefore, in this study, we stratified the T2D patients with five risk factors (BMI ≤25 kg/m$^2$, no physical activity, falls in the last year, HbA1c≥47.5 mmol/mol, and antidiabetic medication treatment), and found that the observed effect size of type 2 diabetes on the risk of fracture decreased when the T2D-related risk factors decreased, and the association became not significant if the type 2 diabetes patients carried none of the risk factors. Unfortunately, we were unable to identify other significant confounding factors related to T2D for fracture. It was supposed that the observed effect of T2D on fracture risk should turn out to be protective when all significant confounding factors were stratified, just like the genetic analysis results. Anyway, the diabetic bone paradox might not exist if the T2D-related risk factors were eliminated. In a recent large-scale cohort study, four factors (duration of T2D, low physical activity, BMI, and insulin treatment) were identified as the important risk factors for fracture among the T2D patients, and the patients without the risk factors had lower fracture risk than their matched controls (*Axelsson, 2023*). One previous study using the UK primary care data found no evidence to suggest a higher risk of fracture in type 2 diabetes patients (*Davie et al., 2021*) and reported that significantly lower fracture risk was observed for overweight individuals (BMI 25–30 kg/m$^2$) in type 2 diabetes than their counterparts without type 2 diabetes (*Davie et al., 2021*).

In summary, by alleviating the bias of unknown confounding factors, we found that the genetically determined T2D did not show "bone paradox". And the shared genetic architecture between T2D and fracture suggested a top signal near *RSPO3* gene. In addition, the stratified prospective regression analysis suggested that the effect size of T2D on the risk of fracture decreased if T2D-related risk factors could be eliminated. Therefore, it is important to manage the complications of T2D to prevent the risk of fracture.

## Methods
### Study participants and wGRS analysis
The UK Biobank data, the application 41376 as we used before (*Bai et al., 2020*), was applied in this study under a prospective design. We identified the individuals with T2D and fracture using the ICD codes and self-report status. The detailed information on the field ID and codes for data extraction from UK Biobank was listed in *Supplementary file 1n*. We excluded participants if they were identified as follows: (1) ethnically identified as non-European (n=30,481); (2) diagnosed as type 1 diabetes (n=4455); (3) diagnosed with diseases associated with bone loss (n=21,560); (4) diagnosed as a fracture with known primary diseases (n=7222) (*Supplementary file 1o*). For the remaining 439,982 samples, we further excluded 145,411 participants with relatedness (kinship) with others in the wGRS analysis (294,571 participants left) (*Figure 1*). In addition, we classified the fracture into weight-bearing bones (neck, vertebrae, pelvic, femur, tibia) (N of fracture = 8992, N of non-fracture=285,579) and other bones (skull and facial, ribs, sternum, forearm, wrist and hand, foot, and other unspecified body regions) (N of fracture = 20,317, N of non-fracture=274,254) using the ICD codes and self-report status (*Supplementary file 1p*).

The summary-statistic data for the type 2 diabetes were obtained from a very recent GWAS consisting of 80,154 individuals with type 2 diabetes and 853,816 controls in the European population (with 10,454,875 SNPs)(*Mahajan et al., 2022*). We drew a set of independent genetic variants with genome-wide significance (p<5 × 10$^{-8}$) from the type 2 diabetes summary-statistic data by LD clumping based on r$^2$ <0.1 in 500 kb window to serve as instrumental variables (n=404) (*Supplementary file 1a*). We constructed the wGRS for the individuals in the UK biobank (294,571 samples with genotypes) as a linear combination of the selected SNPs weighted by their β coefficients on type 2 diabetes: wGRS = β$_1$×SNP$_1$ + β$_2$×SNP$_2$ + … + βn×SNPn. n is the number of instrumental variables

(here n=404 after LD clumping based on $r^2$ <0.1 in 500 kb window). Next, cox proportional hazards regression and linear regression analyses were performed to analyze the association between the wGRS and fracture/BMD adjusted age, sex, BMI, physical activity, fall history, HbA1c, and medication treatments. Besides, regression modeling was used to estimate the effect of gene-environment interaction (T2D *wGRS × sex*) on fracture risk and BMD. In addition to the T2D *wGRS × sex* interaction term, the model was adjusted for covariates: age, sex, BMI, physical activity, fall history, HbA1c, and medication treatments.

## Two-sample MR analyses

To validate the wGRS results, we also performed the two-sample MR analyses that are independent of UK Biobank samples. The summary-statistic data for fracture (a discovery set of 37,857 fracture cases and 227,116 controls with 2,539,800 SNPs) (*Trajanoska et al., 2018*) and BMD (426,824 samples and 13,753,401 SNPs) (*Morris et al., 2019*) were extracted from the GEFOS consortium (http://www.gefos.org/), while the summary data for type 2 diabetes (*Mahajan et al., 2022*) is the same as used in wGRS analysis.

We used the IVW (*Burgess et al., 2013*), simple median and MR-PRESSO (*Verbanck et al., 2018*) approaches in two-sample MR analyses. For the outcome of fracture, we merged the two summary datasets for T2D and fracture (*Mahajan et al., 2022*; *Trajanoska et al., 2018*) and got 2,479,475 overlapping SNPs, of which 6946 SNPs were genome-wide significant for type 2 diabetes. After LD clumping based on $r^2$ <0.1 in 500 kb window, 298 independent genetic variants were left (*Supplementary file 1b*). Similarly, we got 9,204,694 overlapping SNPs for type 2 diabetes and BMD, and 389 independent genetic variants were left after LD clumping (*Supplementary file 1d*). After harmonizing the effects so that they reflect the same allele, 289 (for fracture) and 380 (for BMD) SNPs were finally used in the IVW and simple median MR analysis. Because the presence of horizontal pleiotropy could bias the MR estimates, we additionally used the MR-PRESSO. The two-sample MR analyses were conducted in R version 4.0.2 using TwoSampleMR (*Hemani et al., 2018*), MendelianRandomization (*Yavorska and Burgess, 2017*) and MR-PRESSO (*Verbanck et al., 2018*) packages. Moreover, we regressed the effects of these 298 SNPs of both traits to highlight the overall effect of T2D on fracture with 'grs.summary' function in the R package 'gtx' (http://www2.uaem.mx/r-mirror/web/packages/gtx/gtx.pdf).

## Multivariable Mendelian randomization (MVMR) analysis

Next, we conducted multivariable MR analysis (*Sanderson et al., 2019*; *Sanderson et al., 2021a*) to examine the direct effect of T2D on fracture and BMD adjusted for BMI with 'MVMR' R package (*Sanderson et al., 2021b*). After adjusting for confounders, the effect of exposure on the outcome was considered to be a direct effect. Specifically, we first extracted the overlapping SNPs from the summary data for T2D (*Mahajan et al., 2022*), BMI (*Locke et al., 2015*), and fracture (*Trajanoska et al., 2018*). Then the independent significant SNPs (p<5 × $10^{-8}$ and $R^2$ <0.1) for either T2D or BMI were pooled as instruments. Additionally, we performed SNP harmonization to correct the orientation of alleles. The final IVs used in MVMR were presented in *Supplementary file 1q*.

## Infer the shared genetics

With the summary-statistic GWAS data of type 2 diabetes (*Mahajan et al., 2022*), BMD (*Morris et al., 2019*), and fracture (*Trajanoska et al., 2018*), we performed genome-wide genetic correlation analysis between type 2 diabetes and fracture/BMD by using linkage disequilibrium score regression (LDSC) (*Bulik-Sullivan et al., 2015a*; *Bulik-Sullivan et al., 2015b*) which estimates the degree of shared genetic factors between two traits, We used MiXeR (*Frei et al., 2019a*) to quantify polygenic overlap (e.g. how many unique and shared polygenic SNPs for type 2 diabetes and fracture) irrespective of genetic correlation (*Frei et al., 2019b*). MiXeR models additive genetic effects as a mixture of four components, representing null SNPs in both traits ($\pi_0$); SNPs with a specific effect on the first and on the second trait ($\pi_1$ and $\pi_2$, respectively); and SNPs with non-zero effect on both traits ($\pi_{12}$). The dice coefficient of two traits was estimated as $\frac{2\pi_{12}}{\pi_1+\pi_2+2\pi_{12}}$ (*Frei et al., 2019b*). We constructed conditional quantile-quantile (QQ) plots which reveals the distribution of p values for fracture/BMD conditioning on the significance of association with type 2 diabetes at the level of p<0.1, p<0.01, and p<0.001 to visualize polygenic enrichment (*Schwartzman and Lin, 2011*). We used the ccFDR

approach (*Andreassen et al., 2013a*) to identify the specific shared loci (*Andreassen et al., 2013b*). We used the conditional false discovery rate (condFDR) to detect SNPs associated with fracture-given associations with type 2 diabetes. We denoted condFDR for fracture-given associations with type 2 diabetes as condFDR(fracture|T2D) and *vice versa*, and considered the significance cutoff <0.01. We used conjFDR to identify SNPs jointly associated with type 2 diabetes and fracture. After repeating the condFDR procedure for both traits, the conjFDR analysis reported the loci that exceed a condFDR significance threshold for two traits simultaneously (the maximum between the condFDRs for both traits), conjFDR <0.05 was set as the significance.

We employed the Summary-data-based Mendelian randomization (SMR) method developed by colleagues (*Zhu et al., 2016*) to test the association of the expression level of gene *RSPO3* with BMI (*Locke et al., 2015*), waist circumference, waist-hip ratio (*Shungin et al., 2015*) and MRI-derived visceral adipose (*Agrawal et al., 2022*), type 2 diabetes and fracture using summary-level data from GWAS and expression quantitative trait loci (eQTL) data of the subcutaneous adipose tissue (9,962,255 SNPs included) from the GTEx database (release v8) (https://www.gtexportal.org/home/)(*Consortium and Laboratory, 2017*). In the SMR analysis, the top cis-eQTL genetic variants were used as the instrumental variables (IVs) for gene expression. Additionally, we downloaded the cis-pQTL summary data for the circulating RSPO3 reported in the study by *Sun et al., 2018*, and performed MR analyses to determine the association of circulating RSPO3 with type 2 diabetes and fracture risk using the IVW MR approach.

## Observational analyses

For the 439,982 UK biobank samples (see foregoing description of study participants), we only focused on the participants diagnosed with T2D within the 10 year period from 1 January 2006 to 31 December 2015, leaving 425,772 participants (with 14,860 type 2 diabetes patients) (*Figure 1*). Here, each type 2 diabetes patient had a diagnosis date, taking this date as the reference date, we first calculated the onset age, then among the participants who were free of T2D, we selected up to 27 participants (whenever possible) whose age at the reference date (±3 years) could be matching to the onset age as referents (*Figure 1—figure supplement 1*). In total, 363,884 non-T2D referents were individually matched with a 6 year age band at the reference date. We prospectively followed these type 2 diabetes patients and referents from the reference date until the diagnosis of fracture, death, emigration, 19 April 2021 (diagnose a fracture of the last person in the cohort), whichever came first. Survival time was calculated based on whether the patient had a fracture. If individuals had a fracture, the survival time is calculated as the time of the first diagnosis of fracture minus the reference date. If individuals did not have a fracture, it was defined as the minimum time of the reference date to diagnose a fracture of the last person in the cohort (19 April 2021), death, or emigration date. We excluded 25,865 participants with fracture diagnosis date, or death or emigration date before the reference date, leaving 352,879 participants included in the final analysis (13,817 type 2 diabetes patients and 339,062 referents) (*Figure 1* and *Supplementary file 1*). Cox proportional hazards regression, as a statistical method to analyze the effect of risk factors on the time it takes for a specific event to happen, was used to test the relationship between T2D and fracture. Meanwhile, multiple linear regression analyses were performed to test the association between T2D and BMD. Here, the BMD was estimated from quantitative ultrasound measurement at the heel. The use of the device generates two variables, including speed of sound (SOS) and BUA (the slope between the attenuation of the sound signal and its frequency as it travels through the bone and soft tissue). Heel BMD was calculated by the following formula: BMD = 0.002592 × (BUA +SOS)−3.687.

First, we adjusted for clinical risk factors including reference age, sex, BMI, physical activity, fall history, HbA1c, and medication treatments to examine the relationship between T2D and fracture/BMD (Model 0). To examine the intermediary effect of risk factors on the relationship between T2D and fracture, the mediation analysis was performed using the R packages of 'mediation.' Individuals treated with any glucose-lowering medication including insulin product, metformin, troglitazone, pioglitazone, rosiglitazone, tolbutamide, glibenclamide, gliclazide, glipizide, gliquidone, glimepiride, chlorpropamide, tolbutamide, repaglinide and nateglinide were recorded as having received medical treatment. We also included BMD as an additional confounding factor for fracture analysis as a complement to the basic model (Model 1). In addition, we classified the fracture into weight-bearing bones (neck, vertebrae, pelvic, femur, tibia) (N of fracture = 8992, N of non-fracture=285,579) and

other bones (skull and facial, ribs, sternum, forearm, wrist and hand, foot and other unspecified body regions) (N of fracture = 20,317, N of non-fracture=274,254) using the ICD codes and self-report status (*Supplementary file 1p*). As we did in wGRS analysis, we classified the fracture into weight-bearing bones and other bones fracture. Briefly, 6582 (1.92%) participants were identified as weight-bearing bones and 9586 (2.77%) participants were identified as other bones. Second, we carried out stratified analyses between type 2 diabetes, fracture, and BMD based on the T2D-related risk factors for fracture. We took five clinical factors to classify the individuals at risk, for example, if an individual had BMI ≤25 kg/m$^2$, no physical activity, falls in the last year, HbA1c≥47.5 mmol/mol, and antidiabetic medication treatment, this individual was identified to have five risk factors, and so forth. Based on the number of risk factors, we grouped 13,817 individuals with T2D into subgroup for analysis. This analysis was adjusted for reference age and sex. For the BMD analysis, the age when attended the assessment center was included in the analysis instead of the reference age.

## Acknowledgements

We thankfully acknowledge the High-performance Computing Center at Westlake.

## Additional information

### Funding

| Funder | Grant reference number | Author |
| --- | --- | --- |
| The "Pioneer" and "Leading Goose" R&D Program of Zhejiang | 2023C03164 | Hou-Feng Zheng |
| National Natural Science Foundation of China | 82370887 | Hou-Feng Zheng |
| The Chinese National Key Technology R&D Program, Ministry of Science and Technology | 2021YFC2501702 | Hou-Feng Zheng |
| The "Pioneer" and "Leading Goose" R&D Program of Zhejiang | 2024SSYS0032 | Hou-Feng Zheng |
| The Westlake Laboratory of Life Sciences and Biomedicine | 202208014 | Hou-Feng Zheng |

The funders had no role in study design, data collection and interpretation, or the decision to submit the work for publication.

### Author contributions

Pianpian Zhao, Data curation, Formal analysis, Validation, Investigation, Visualization, Methodology, Writing – original draft, Writing – review and editing; Zhifeng Sheng, Writing – original draft, clinical interpretation of the results; Lin Xu, Writing – original draft, clinical interpretation of the results; Peng Li, Wenjin Xiao, Chengda Yuan, Zhanwei Xu, Mengyuan Yang, Yu Qian, Jiadong Zhong, Jiaxuan Gu, David Karasik, Data curation, Writing – review and editing; Hou-Feng Zheng, Conceptualization, Resources, Supervision, Funding acquisition, Project administration, Writing – review and editing

### Author ORCIDs

Pianpian Zhao ![ORCID] https://orcid.org/0009-0009-1676-1748
Hou-Feng Zheng ![ORCID] http://orcid.org/0000-0001-5681-8598

### Ethics

All individuals provided written informed consent. The North West Multi-Centre Research Ethics Committee approved the UK Biobank ethical application (reference number: 16/NW/0274).

Reviewer #1 (Public review): https://doi.org/10.7554/eLife.89281.3.sa1
Reviewer #2 (Public review): https://doi.org/10.7554/eLife.89281.3.sa2
Author response https://doi.org/10.7554/eLife.89281.3.sa3

## Additional files

### Supplementary files

• MDAR checklist

• Supplementary file 1. Supplementary tables. (**a**) Characteristics of the 404 single nucleotide polymorphisms associated with type 2 diabetes. (**b**) The information of 298 lead SNPs for type 2 diabetes with the risk of fracture. (**c**) Multivariable MR analysis of the direct effect of BMI on fracture risk and BMD. (**d**) The information of 389 SNPs for type 2 diabetes with BMD. (**e**) Summary of pariwise genetic correlation using linkage disequilibrium score regeression(LDSC). Abbreviations: p1 = trait 1, p2 = trait 2, rg = genetic correlation, se = standard error of rg, p = p-value for rg, gcov_int, gcov_int_se = cross-trait LD Score regression intercept and standard error. (**f**) The results of cross-trait analysis with MiXeR model for type 2 diabetes, fracture and BMD. Abbreviations: nc1@p9, nc2@p9, and nc12@p9 = the number of causal variants for trait1, trait2 and both, respectively; nc@p9 = number of trait-influencing variants specific to type 2 diabetes; rho_beta = the correlation of effect sizes within the shared polygenic component; rg = genetic correlation; concordant_fraction = the proportion of shared variants with concordant direction of effect in both traits on all shared variants; pi1@p9, pi2@p9, and pi12@p9 = polygenicity of trait1, trait2 and both, respectively; The best_vs_min_AIC and the best_vs_max_AIC indicates whether MiXeR can accurately distinguish the reported overlap from the minimum and maximum possible overlap allowed, respectively. (**g**) Distinct genomic loci shared between type 2 diabetes and fracture at conjFDR<0.05. (**h**) The results of SMR analysis for the expression of RSPO3 (ENSG00000146374) with type 2 diabetes, fracture, BMI, WC, WHR and VAT in adipose subcutaneous. (**i**) Distinct genomic loci shared between type 2 diabetes and BMD at conjFDR<0.05. (**j**) The characteristics of participants and comparison between individuals with and without type 2 diabetes. (**k**) The regression between type 2 diabetes, fracture and BMD. (**l**) Assessment of the mediators (BMI) for the association between type 2 diabetes and fracture. (**m**) Baseline characteristics of participants with different Numbers of Risk Factors. (**n**) Detailed information on the field ID and codes for participants included in UK Biobank. Abbreviations: ICD-9, the International Classification of Diseases, 9th Revision; ICD-10, the International Classification of Diseases, 10th Revision. (**o**) The International Classification of Diseases (ICD) codes and self-reported codes for excluded diseases. (**p**) Detailed information on the field ID and codes for specific fracture sites included in UK Biobank. (**q**) Detailed information for instrumental variables of T2D and BMI on fracture and BMD.

### Data availability

All data generated during this study are included in the manuscript. Summary-level analysis was conducted using publicly available data as described below. Data for fracture was extracted from *Trajanoska et al., 2018*, and downloaded from the GEnetic Factors for OSteoporosis (GEFOS) website. Data for BMD GWAS was extracted from *Morris et al., 2019*, and downloaded from the same GEFOS website. T2D data was obtained from the DIAbetes Genetics Replication And Meta-analysis (DIAGRAM) Consortium under "Ancestry specific GWAS meta-analysis summary statistics: European" published in *Mahajan et al., 2022*. Data for BMI, WC and WHR GWAS was downloaded from the Genetic Investigation of ANthropometric Traits (GIANT) Consortium at https://portals.broadinstitute.org/collaboration/giant/index.php/GIANT_consortium_data_files, under "GWAS Anthropometric 2015 BMI Summary Statistics (subtitle: Download BMI EUR Ancestry GZIP)" published in *Locke et al., 2015*, and under "GWAS Anthropometric 2015 Waist Summary Statistics (subtitle: WC: Download WC Combined EUR GZIP and WHR: Download WHR Combined EUR GZIP)" published in *Shungin et al., 2015*. Data for MRI-derived visceral adipose could be downloaded from the Cardiovascular Disease Knowledge Portal (CADKP), under "Fat distribution 2022 GWAS" published in *Agrawal et al., 2022*. Data for eQTL data could be obtained from Genotype-Tissue Expression Portal (GTEx_Analysis_v8_eQTL). Additionally, summary association results for the cis-pQTL were downloaded from *Sun et al., 2018*, which is available at https://wbbc.westlake.edu.cn/downloads_proteins.html

(RSPO3.13094.75.3). Please note that individual-level genetic and phenotype data require permission from the UK Biobank with accession ID 41376.

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
