## [Editor Report · eLife assessment]

This study aims to explore the diabetes-bone paradox using the Mendelian Randomization approach. That diabetes itself is not the direct cause, but rather the complications or associated risk factors increase the risk of fracture, constitutes a **valuable** insight. Mendelian randomization to explain the relationship of two complex conditions is **solid** and conducted properly; however, the efforts to reconcile the discrepancies between the Mendelian Randomization analysis and observational studies are **incomplete**.

---

## [Referee Report · Reviewer #1 (Public review)]

Summary:

The manuscript of Zhao et al. aimed at investigating the relationships between type 2 diabetes, bone mineral density (BMD) and fracture risk using Mendelian Randomization (MR) approach.

The authors found that genetically predicted T2D was associated with higher BMD and lower risk of fracture, and suggested a mediated effect of RSPO3 level. Moreover, when stratified by the risk factors secondary to T2D, they observed that the effect of T2D on the risk of fracture decreased when the number of risk factors secondary to T2D decreased.

Strengths:

- Important question

- Manuscript is overall clear and well-written

- MR analyses have been conducted properly, which include the usage of various MR methods and sensitivity analyses, and likely meet the criteria of the MR-strobe checklist to report MR results.

Weaknesses:

- Interpretation of MR findings should be more nuanced given the modest (almost neutral) relationship between T2D and fracture risk in MR

---

## [Referee Report · Reviewer #2 (Public review)]

The authors employed the Mendelian Randomization method to analyze the association between type 2 diabetes (T2D) and fracture using the UK Biobank data. They found that "genetically predicted T2D was associated with higher BMD and lower risk of fracture". Additionally, they identified 10 loci that were associated with both T2D and fracture risk, with the SNP rs4580892 showing the highest signal. While the negative relationship between T2D and fracture has been previously observed, the discovery of these 10 loci adds an intriguing dimension to the findings, although the clinical implications remain uncertain.

Many thanks for your response which has clarified my understanding of your paper. And, thank you for the additional analyses. I still find the paper challenging to understand due to two different analyses that yielded conflicting results: (a) in the observational analysis, the authors found that type 2 diabetes was associated with both higher BMD and a higher risk of fracture (ie a paradox); but (b) in the Mendelian randomization analysis, 'genetically predicted type 2 diabetes' was associated with greater BMD and a lower risk of fracture. I consider that your conclusion is not consistent with the data you presented.

---

## [Author Response]

The following is the authors’ response to the original reviews.

**Reviewer #1 (Public review):**
Summary:The manuscript of Zhao et al. aimed at investigating the relationships between type 2 diabetes, bone mineral density (BMD) and fracture risk using Mendelian Randomization (MR) approach.The authors found that genetically predicted T2D was associated with higher BMD and lower risk of fracture, and suggested a mediated effect of RSPO3 level. Moreover, when stratified by the risk factors secondary to T2D, they observed that the effect of T2D on the risk of fracture decreased when the number of risk factors secondary to T2D decreased.Strengths:Important questionManuscript is overall clear and well-writtenMR analyses have been conducted properly, which include the usage of various MR methods and sensitivity analyses, and likely meet the criteria of the MR-strobe checklist to report MR results.

Response: Thanks.

Weaknesses:Previous MR studies on that topic have not been discussed

Response: In the manuscript, we discussed the previous MR findings from Trajanoska et al., BMJ, 2018. This study assessed the effect of 15 clinical risk factors (including type 2 diabetes) on fracture risk. Now we have included the other two studies (Mitchell et al, Diabetologia, 2021; Ahmad et al JBMR, 2016) which took BMD as the exposure in the paragraph when we discussed the effects on BMD.

Multivariable MR could have been used to better assessed the mediative effect of BMI or RSPO3 on the relationships between T2D and fracture risk.

Response: In revision, the inverse weighted multivariable MR model was used to estimate the direct effect of T2D upon the fracture and BMD adjusted for BMI with ‘MVMR’ R package (https://github.com/WSpiller/MVMR). Specifically, we first extracted the overlapping SNPs from the summary data for T2D, BMI and fracture. Then the independent significant SNPs (P<5×10−8 and R2<0.1) for either T2D or BMI were pooled as instruments. Additionally, we performed SNP harmonization to correct the orientation of alleles. Additionally, we performed SNP harmonization to correct the orientation of alleles. The results showed that increased risk of T2D has a direct effect that decreased fracture risk (OR=0.974, 95%CI=0.953-0.995, P=0.017 adjusted BMI), and BMI mediated 9.03% of the protective effect. The multivariable MR analysis suggested that T2D also showed direct effect on increased BMD after adjusting for BMI (β=0.042, 95%CI=0.026-0.057, P=1.92×10-7). We didn’t observe the direct effect of MRI-derived visceral (β=0.02, P=0.831) and abdominal subcutaneous (β=0.03, P=0.57) on fracture risk adjusted for RSPO3 expression. We have updated the Methods and Results accordingly.

**Reviewer #2 (Public review):**
The authors employed the Mendelian Randomization method to analyze the association between type 2 diabetes (T2D) and fracture using the UK Biobank data. They found that "genetically predicted T2D was associated with higher BMD and lower risk of fracture". Additionally, they identified 10 loci that were associated with both T2D and fracture risk, with the SNP rs4580892 showing the highest signal. While the negative relationship between T2D and fracture has been previously observed, the discovery of these 10 loci adds an intriguing dimension to the findings, although the clinical implications remain uncertain.

Response: We appreciate the reviewer's thoughtful evaluation of our study. The hypothesis and idea of this study is that the genetically determined type 2 diabetes might not be associated with higher risk of fracture, but the risk association could be observed. However, when stratified by the risk factors secondary to the disease, we observed that the effect of T2D on the risk of fracture decreased when the number of risk factors secondary to T2D decreased, and the association became non-significant if the T2D patients carried none of the risk factors. These results suggested that the risk factors secondary to type 2 diabetes might contribute more to the risk of fracture. Therefore, the clinical implications of our study might lie in the health management of type 2 diabetes patients. We suggest that it is important to manage the complications of type 2 diabetes to prevent the risk of fracture.

**Reviewer #1 (Recommendations for the authors):**
Introduction/discussion: findings from MR previously published on that topic have not been discussed in this manuscript (eg, Mitchell et al, Diabetologia, 2021; Ahmad et al JBMR, 2016);

Response: In the manuscript, we discussed the previous MR findings from Trajanoska et al., BMJ, 2018. The study assessed the effect of 15 clinical risk factors (including type 2 diabetes) on fracture risk. Sorry that we missed the studies you mentioned, these two studies took BMD as the exposure, now we have included them in the paragraph where we discussed the effect of T2D on BMD (Page 14, Line 320-322).

In the one-sample MR analysis: I would suggest looking at whether the association between T2D GRS and fracture risk differ across fracture sites; in the hypothesis that BMI might be protective, performing the analysis separately for weight-bearing bones vs not weight-bearing bones would be interesting.

Response: According to your suggestion, we further categorized fractures into weight-bearing bones (neck, vertebrae, pelvic, femur, tibia) and other bones (detailed codes have been added to Supplementary Table 16). When we regressed the observed fracture on the wGRS, it indicated that there was trend of protective association between T2D wGRS and both weight-bearing bones fracture (OR=0.9772, 95%CI=0.9552-0.9997, P=0.04737, N of fracture=8,992) and other bones fracture (OR=0.9838, 95%CI=0.9688-0.9991, P=0.0386, N of fracture=20,317) (Figure 1). We have updated the Methods and Results accordingly (Page 6, line 129-134 and Page 18, line 408-412).

In this analysis, I would also suggest verifying the absence of sex interaction with T2D PRS on BMD and fracture risk

Response: Thanks for your suggestion, we further estimated the effect of sex interaction on BMD and fracture risk with T2D wGRS × sex interaction term in regression model. And you are right, we found no interactions (sex with T2D wGRS) on fracture risk (P=0.5576) and BMD (P=0.66). Moreover, we conducted the stratified analysis by sex. When we regressed the observed fracture on the wGRS in male, we found that the genetically determined type 2 diabetes was also associated with lower risk of fracture (OR=0.977, P=0.015) (adjusting for reference age, sex, BMI, physical activity, fall history, HbA1c and medication treatments). In female, the direction of the association remained with no significance (OR=0.986, P=0.139). We tested the heterogeneity between male and female, and found no significant difference (Pheterogeneity=0.457). Similarly, the genetically determined type 2 diabetes was associated with higher BMD in male (β=0.023, P=8.23×10-14) and female (β=0.022, P<2.0×10-16), and Pheterogeneity=0.6306 (Supplementary Figure 2). We have updated the Methods and Results accordingly (Page 6, line 134-139 and Page 19, line 425-429).

In the two-sample MR analysis: I would suggest performing a multivariable MR to look at the effect of T2D adjusted for BMI on BMD and fracture risk (see Burgess et al, AJE, 2016)

Response: Thanks for your suggestion, in revision, the inverse weighted multivariable MR model was used to estimate the direct effect of T2D upon the fracture and BMD adjusted for BMI with ‘MVMR’ R package (https://github.com/WSpiller/MVMR). Specifically, we first extracted the overlapping SNPs from the summary data for T2D, BMI and fracture. Then the independent significant SNPs (P<5×10−8 and R2<0.1) for either T2D or BMI were pooled as instruments. Additionally, we performed SNP harmonization to correct the orientation of alleles. Additionally, we performed SNP harmonization to correct the orientation of alleles. The final IVs used in MVMR were presented in Supplementary Table 17. The results showed that increased risk of T2D has a direct effect that decreased fracture risk (OR=0.974, 95%CI=0.953-0.995, P=0.017 adjusted BMI) and increased BMD (β=0.042, 95%CI=0.026-0.057, P=1.92×10-7 adjusted BMI). We have updated the Methods and Results accordingly (Page 7, line 155-158, 162-164, and Page 20, line 456-465).

In the section "infer the shared genetics". In addition of using waist circumference and waist-hip ratio, it would have been interesting to use GWAS summary statistics for subcutaneous and visceral adiposity (Agrawal, Nat Comm, 2022), and look at through multivariable MR whether RSPO3 mediate the effect of subcutaneous fat on fracture risk.

Response: Thanks for your suggestion, we downloaded the genetic summary data from Agrawal, Nat Comm, 2022, and performed the same SMR analysis as we did before. We found that higher expression of RSPO3 was associated with higher MRI-derived visceral (β=0.199, P=4.36×10-5). We have updated the Methods and Results accordingly (Page 9, line 206-208 and Page 22, line 494-495).

We didn’t observe the direct effect of MRI-derived visceral (β=0.02, P=0.831) and abdominal subcutaneous (β=0.03, P=0.57) on fracture risk adjusted for RSPO3 expression.

**Reviewer #2 (Recommendations for the authors):**
Specific commentsSeveral concerns regarding the study's concept and methodology should be addressed before accepting the findings as credible. I would like to invite the authors to comment on the following points.(1) I find the authors' assertion that individuals with type 2 diabetes (T2D) exhibit both higher BMD and an increased risk of fracture to be unconvincing. The BMD measurement they refer to is based on areal BMD, which fails to account for the three-dimensional aspect of bone density. Existing evidence suggests that patients with T2D actually have lower trabecular bone scores (a predictor of fracture risk) compared to those without the condition. Furthermore, there is a lack of a clearly stated hypothesis underlying the study.

Response: Yes, in this study, the bone mineral density measurement is based on areal BMD. We made this clear in Abstract. And we agree that other measurements, such as trabecular bone score and chest CT texture analysis, could provide additional valuable information in the evaluation of fracture risk, especially in type 2 diabetes patients. We have discussed this in the manuscript (Page 13, line 295-300). Epidemiologic studies from the past decade provided evidence that increased bone fracture risk is one of the complications of type 2 diabetes. but the areal BMD in type 2 diabetes patients could be normal or even higher (Botella Martinez et al., 2016; Romero-Diaz et al., 2021).

In this study, we employed the mendelian randomization approach to investigate the relationship between type 2 diabetes and fracture/BMD, this method might facilitate the use of genetic data as instrumental variables to alleviate the bias of the unknown confounding factors. We found that the genetically predicted type 2 diabetes was associated with higher BMD and lower risk of fracture. That is to say, by alleviating the bias of the unknown confounding factors through MR analysis, the genetically predicted type 2 diabetes did not show bone paradox.

We then performed observational analysis in UK Biobank, and found that type 2 diabetes was associated with higher risk of fracture and increased BMD. Further, we stratified the T2D patients with five secondary risk factors (BMI≤25kg/m2, no physical activity, falls in the last year, HbA1c≥47.5mmol/mol and antidiabetic medication treatment), and found that the effect of type 2 diabetes on the risk of fracture decreased when the risk factors secondary to type 2 diabetes decreased, and the association became not significant if the type 2 diabetes patients carried none of the risk factors. That is to say, the diabetic bone paradox might not exist if the secondary risk factors of type 2 diabetes were eliminated.

The hypothesis and idea we want to deliver is that the genetically determined type 2 diabetes might not be associated with higher risk of fracture, but the association could be observed. However, when stratified by the risk factors secondary to the disease, we observed that the effect of T2D on the risk of fracture decreased when the number of risk factors secondary to T2D decreased, and the association became non-significant if the T2D patients carried none of the risk factors. These results suggested that the risk factors secondary to type 2 diabetes might contribute more to the risk of fracture. Therefore, it is important to manage the complications of type 2 diabetes to prevent the risk of fracture.

In addition, although we observed type 2 diabetes was observed to be associated with higher risk of fracture, but BMI mediated 30.2% of the protective effect. And the shared genetic architecture between type 2 diabetes and fracture suggested a top signal near RSPO3 gene. Higher expression of RSPO3 was associated with higher waist circumference and higher waist-hip ratio. These results suggested that relatively higher BMI in type 2 diabetes patients might benefit the higher BMD, as our previous study suggested that keeping moderate-high BMI (overweight) might be of benefit to old people in terms of fracture risk(Zhu et al., 2022).

(2) It is not a good idea to solely concentrate on overall fracture risk as it may obscure the potential relationship between T2D and specific fracture sites, such as hip and vertebral fractures. By solely considering total fracture incidence, important associations at individual fracture sites could be overlooked. I would like to propose that the authors expand their analysis to include the examination of hip and vertebral fractures. By incorporating these specific fracture types into their study, a more comprehensive understanding of the association between T2D and fractures can be achieved.

Response: This is a good suggestion, incorporating with the comments from another reviewer, and considering the sample size, we classified fractures into weight-bearing fractures (neck, vertebrae, pelvic, femur, tibia) and other bones (skull and facial, ribs, sternum, forearm, wrist and hand, foot and other unspecified body regions) fracture. We identified 6,582 (1.87%) participants with weight-bearing bones fracture and 9,586 (2.72%) participants with other bones fracture within the 352,879 UK Biobank participants. We observed a higher risk of fracture in the type 2 diabetes patients in the cox proportional hazards regression after adjusted for the reference age, sex, BMI, physical activity, fall history, HbA1c and medication treatments (weight-bearing bones fracture: HR=1.792, 95%CI 1.555-2.065, P=8.25×10-16; other bones fracture: HR=1.337, 95%CI 1.167-1.531, P=2.85×10−5), and additionally controlled for BMD (weight-bearing bones fracture: HR=1.850, 95%CI 1.602-2.136, P<2×10−16; other bones fracture: HR=1.377, 95%CI 1.199-1.580, P=5.54×10−6). We have updated the manuscript according in Results, Methods and Figures (Page 11, line 245-250; Page 24, line 540-547; Figure 4A).

(3) I consider that there is an issue with combining data from both males and females in the analysis. It is widely recognized that women generally have a higher risk of fracture compared to men. Moreover, the association between BMD and fracture may vary between genders, and the risk of T2D is typically lower in women than in men. Therefore, I strongly recommend that the analysis be stratified by gender to account for these differences and provide a more accurate understanding of the relationships involved.

Response: Thanks for your suggestion, we now add the stratified results by sex to each analysis. Briefly, in wGRS analysis, we found that the genetically determined type 2 diabetes was associated with lower risk of fracture in male (OR=0.977, 95%CI=0.958-0.995, P=0.015) (adjusting for reference age, sex, BMI, physical activity, fall history, HbA1c and medication treatments). The association in female was not significant, but the direction is the same as the male (OR=0.986, 95%CI=0.969-1.004, P=0.139). Meanwhile, the genetically determined type 2 diabetes was associated with higher BMD in both male (β=0.023, 95%CI=0.017-0.030, P=8.23×10−14) and female (β=0.022, 95%CI=0.017-0.026, P<2×10−16). In observational analysis, we observed a higher risk of fracture in the type 2 diabetes patients in the cox proportional hazards regression after adjusted for the reference age, sex, BMI, physical activity, fall history, HbA1c and medication treatments in male (HR=1.587, 95%CI 1.379-1.828, P=1.26×10−10) and female (HR=1.530, 95%CI 1.334-1.756, P=1.27×10−9), respectively. When we additionally controlled for BMD (HR=1.607, 95%CI 1.393-1.853, P=7.21×10−11 in male; HR=1.601, 95%CI 1.393-1.841, P=3.59×10−11 in female), we still observed increased risk of fracture in type 2 diabetes (Page 6, line 136-139; Page 11, line 241-243).

(4) My understanding is that "BMD" in UK Biobank refers to estimated BMD derived from ultrasound measurements, rather than being directly measured using dual-energy X-ray absorptiometry (DXA). It would be helpful to clarify whether the BMD mentioned in the manuscript refers to estimated BMD or DXA-based BMD to ensure accurate interpretation of the results.

Response: Yes, we used the BMD estimated from quantitative ultrasound measurement at heel as the outcome. Use of the device generates two variables, including speed of sound (SOS) and BUA (the slope between the attenuation of the sound signal and its frequency as it travels through the bone and soft tissue). Heel BMD was calculated by the following formula: BMD = 0.002592 ×(BUA+SOS)−3.687. We have made this clear in Methods (Page 23, line 526-530).

(5) The clarification regarding the nature of the 13,817 individuals with T2D mentioned in Supplementary Table 9 is needed. It is unclear whether this figure represents incidence or prevalence. If it refers to incidence, it would be informative to specify the duration of the follow-up period for these individuals.

Response: The UK Biobank data (application #41376), was applied in our study under a prospective design. We excluded participants if they were identified as follows: (1) ethnically identified as non-European (n = 30,481); (2) diagnosed as type 1 diabetes (n=4,455); (3) diagnosed with diseases associated with bone loss (n=21,560); (4) diagnosed as fracture with known primary diseases (n=7,222) (Supplementary Table 15). For the 439,982 UK biobank samples, we focused the participants diagnosed with T2D within the 10-year period from 1 January 2006 to 31 December 2015, leaving 425,772 participants (with 14,860 type 2 diabetes patients). Here, each type 2 diabetes patient had a diagnosis date (i.e., the reference date), we first calculated the onset age, then among the participants who were free of T2D, we selected up to 27 participants (whenever possible) whose age at the reference date (± 3 years) could be matching to the onset age as referents. In total, 363,884 non-T2D referents were individually matched with 6-year age band at the reference date. We prospectively followed these type 2 diabetes patients and referents from the reference date until diagnosis of fracture, death, emigration, 19 April 2021 (diagnose a fracture of the last person in the cohort), whichever came first (with the mean duration of type 2 diabetes 8.34 years). Survival time was calculated based on whether the patient had a fracture. If individuals had a fracture, the survival time is calculated as the time of the first diagnosis of fracture minus the reference date. If individuals did not have a fracture, it was defined as the minimum time of the reference date to diagnose a fracture of the last person in the cohort (19 April 2021), death, or emigration date. We excluded 25,865 participants with fracture diagnosis date, or death or emigration before the reference date, leaving 352,879 participants included in the final analysis (13,817 type 2 diabetes patients and 339,062 referents). We identified 16,147 (4.6%) participants with fracture within the 352,879 UK Biobank participant. We have made this clear in the Methods and Results (Page 18, line 400-406; Page 22-23, line 506-523; Page 10, line 231-233).

(6) I find the selection of participants for the analysis to be highly problematic. Supplementary Figure 1 suggests that individuals with a history of fracture were excluded from the study. However, it is well established that prior fracture history is a significant predictor of future fractures. Therefore, the exclusion of participants with prior fractures likely introduced selection bias into the analysis, potentially compromising the study's findings.

Response: Sorry that we used a misleading term “secondary fracture” in the manuscript and figure. What we want to say here is that “the participants diagnosed as fracture with known primary diseases” (n=7,222), because we want to investigate the effect of diabetes on fracture, we should exclude other factures with known reason. We have changed the term in the manuscript and figure accordingly (Page 18, line 405-406; Supplementary Figure 1).

Since this study is a prospective design, all the participants did not have fracture at the reference date, we prospectively followed these type 2 diabetes patients and referents from the reference date until diagnosis of fracture, death, emigration, 19 April 2021 (diagnose a fracture of the last person in the cohort), whichever came first. Therefore, each study subject either had one fracture or no fracture.

(7) It is unclear what exactly is meant by "genetically predicted T2D." Could it possibly refer to the polygenic risk score derived from the variants associated with T2D? Clarification is needed regarding the methodology used to determine this "genetically predicted T2D" and its relation to the construction of a polygenic risk score based on T2D-associated variants.

Response: In this study, we used weighted genetic risk score (wGRS) method and two-sample Mendelian Randomization (MR) method to estimate the effect of genetically predicted T2D on fracture. We constructed the wGRS for the individuals in the UK biobank (294,571 samples with genotypes) as a linear combination of the selected SNPs weighted by their β coefficients on type 2 diabetes: wGRS = β1 ×SNP1 + β2 ×SNP2 + … + βn ×SNPn. n is the number of instrumental variables. To validate the wGRS results, we also performed the two-sample MR analyses that is independent of UK Biobank samples. We used three two-sample MR approaches, the inverse variance weighting (IVW), simple median and MR-PRESSO approaches. Both methods took the genetically predicted type 2 diabetes as the exposure (See Methods Page 18, line 419-422; Page 19, line 439-440).

(8) My understanding is that the Mendelian Randomization analysis relies on, among others, 2 assumptions: (1) the genetic marker is linked to the exposure (e.g., T2D), and (2) the genetic marker remains independent of the outcome (e.g., fracture) when considering the exposure and all confounding factors. In the authors' study, they identified 10 loci that exhibited associations with both T2D and fracture risk. This finding raises questions about whether the assumptions underlying Mendelian Randomization have been violated?

Response: You're absolutely right. Because the presence of horizontal pleiotropy could bias the MR estimates, we additionally used the MR pleiotropy residual sum and outlier (MR-PRESSO) method. When we excluded pleiotropic variants using restrictive MR-PRESSO method, the causal association was still detected between type 2 diabetes and fracture (OR=0.967, 95%CI=0.945-0.989, P=0.004) (Page 6, line 146-149).

(9) The analysis provided in Supplementary Table 10 appears to have certain limitations. From my understanding, the analysis treated fracture and BMD as outcome variables, with T2D regarded as the predictor variable. However, what is of interest is whether the association between T2D and fracture remains significant even after accounting for well-established risk factors for fractures, including BMD. It is crucial to determine whether the association between T2D and fracture is independent of these established risk factors. Therefore, I suggest the authors consider the following 3 models:Model 1: fracture ~ age + T2DModel 2: fracture ~ age + T2D + BMDModel 3: fracture ~ age + T2D + BMD + fracture history + falls

Response: In our previous analysis, we have adjusted for 7 covariates (including fall history) in the basic model for fracture, i.e.

fracture ~ T2D + age + sex + BMI + physical activity + HbA1c + medication treatments + fall history (Model 0)

We have already included “fall history” in the basic model, according to your suggestion, we further considered an additional model for fracture by including BMD as the covariate:

fracture ~ T2D + age + sex + BMI + physical activity + HbA1c + medication treatments + fall history + BMD (Model 1)

We cannot include fracture history as the covariate because each study subject either had one fracture or no fracture, as we also answered in Question 6.

In model 0, we observed a higher risk of fracture in the type 2 diabetes patients in the cox proportional hazards regression after adjusted for the clinical risk factors including reference age, sex, BMI, physical activity, HbA1c, medication treatments and fall history (HR=1.527, 95%CI=1.385-1.685, P<2.0×10-16). When we additionally controlled for BMD (model 1), we still observed increased risk of fracture in type 2 diabetes (model 1: HR=1.574, 95%CI=1.425-1.739, P<2.0×10-16) (Supplementary Table 11).

We thank for your suggestion, and we have updated accordingly in Methods, Results, and Figures (Page 11, line 243-245; Page 24, line 539-540; Figure 4A).

(11) The dichotomization of data presented in Figure 4 is not considered ideal, as this approach often leads to a loss of valuable information. It is strongly recommended that the authors reconsider their data analysis strategy and reanalyze the data using continuous variables, such as BMI and HbA1c, to capture a more nuanced understanding of the relationships involved.

Response: We agree that dichotomization of data would lead to a loss of valuable information. In model 0 and model 1, we used the continuous variables in the analyses, we adjusted for the reference age, sex, BMI (as a continuous variable), physical activity, fall history, HbA1c (as a continuous variable) and medication treatments to analyze the relationship between type 2 diabetes and fracture in the cox proportional hazards regression. We have updated the Figure 4 accordingly.

In stratified analyses, we took 5 clinical factors secondary to the diseases to classify the individuals at risk, for example, if an individual had BMI≤25kg/m2, no physical activity, falls in the last year, HbA1c≥47.5mmol/mol and antidiabetic medication treatment, this individual was identified to have 5 risk factors, and so forth. Finally, 2,303 patients carried none of the risk factors, 4,128 patients accompanied with one of the risk factors, and 4,252 patients carried at least two risk factors. We found that the effect of type 2 diabetes on the risk of fracture decreased when the risk factors secondary to type 2 diabetes decreased. We have made this clearer in the Methods and Results (Page 11, line 255-257; Page 24, line 548-552).

(12) The conclusion of the study appears to be somewhat confusing. In the Abstract, the authors initially state that "genetically predicted T2D was associated with higher BMD and lower risk of fracture." However, later on, they write that "the genetically determined T2D might not be associated with a higher risk of fracture." This discrepancy raises uncertainty about the clear take-home message of the study.

Response: Here we just want to deliver the same message by different statements, avoiding the repeat of writing. The take-home message we want to deliver is that the genetically determined type 2 diabetes might not be associated with higher risk of fracture, but the association could be observed, suggesting the risk factors secondary to type 2 diabetes might contribute more to the risk of fracture. Therefore, it is important to manage the complications of type 2 diabetes to prevent the risk of fracture, especially the 5 factors we investigated in this study.

(13) Apologies if I offend It seems that the authors lack comprehensive knowledge of the osteoporosis literature. In the Introduction, their definition of osteoporosis as "an age-related common disease characterized by low bone mass" is inadequate. It would be advisable for the authors to provide a more widely accepted and standard definition of osteoporosis to ensure accuracy and alignment with established definitions in the field.

Response: Thanks for your suggestion. Now we changed the statement as follow “Osteoporosis is a common chronic disease characterized by low bone mass and disruption of bone microarchitecture. Fragility fracture is the ultimate outcome of poor bone health”.

(14) There are several instances in which the authors use non-standard terminologies. For example, the use of the word 'effects' (in "the observed effect of T2D on fracture risk") is inappropriate since this study is observational in nature.

Response: In statistics, an effect size is a value measuring the strength of the relationship between two variables in a population. We have changed some of the words “effect” into “effect size” (whenever appropriate) to refer the Hazard ratio between T2D on fracture.

(15) Please provide a reference for "diabetic bone paradox".

Response: We have cited Botella Martínez et al, Endocrinol Nutr. 2016 and Romero-Díaz et al, Diabetes Ther. 2021 in both Introduction and Discussion (Page 3, line 76-77; Page 13, line 295-297).

References

Botella Martinez S, Varo Cenarruzabeitia N, Escalada San Martin J, Calleja Canelas A. The diabetic paradox: Bone mineral density and fracture in type 2 diabetes. Endocrinol Nutr. 2016, 63: 495-501.

Romero-Diaz C, Duarte-Montero D, Gutierrez-Romero SA, Mendivil CO. Diabetes and bone fragility. Diabetes Ther. 2021, 12: 71-86.

Zhu XW, Liu KQ, Yuan CD et al. General and abdominal obesity operate differently as influencing factors of fracture risk in old adults. iScience. 2022, 25: 104466.